# Study of 405 nm Laser-Induced Time-Resolved Photoluminescence Spectroscopy on Spinel and Alexandrite

**Wenxing Xu** [1,*] **, Tsung-Han Tsai** [2,*] **and Aaron Palke** [2]

1  Independent Researcher, Cheerstrasse 13, 6014 Lucerne, Switzerland
2  Gemological Institute of America, 50 W 47th Street, New York, NY 10036, USA
*  Correspondence: wenxingx@gmail.com (W.X.); stsai@gia.edu (T.-H.T.)

**Abstract:** Research on photoluminescence spectroscopy on Cr-doped gem materials has demonstrated great success regarding the identification of gemstones in terms of building rapid test systems. In this study, 405 nm photoluminescence spectroscopy was used to measure the luminescence decay profiles of dozens of natural and lab-grown spinel (including heated spinel) and alexandrite. Spinel and alexandrite are both capable of producing photoluminescence with a long lifetime: spinel between 9 and 23 microseconds and alexandrite from 25 to 53 microseconds. The photoluminescence lifetime and exponential parameters of the half-life demonstrated notable differences in the ranges of decay times between natural, heated, and lab-grown versions of these materials.

**Keywords:** time-resolved spectroscopy; spinel; alexandrite; photoluminescence lifetime; fluorescence decay; gemstone testing





## 1. Introduction

Studies of photoluminescence (PL) spectroscopy on colored gemstones have been successful in identifying different gemstone species such as corundum, spinel, emerald, and alexandrite, which can be used in rapid testing systems [1]. Time-resolved PL spectroscopy can detect events within the environment of a fluorophore. The events that can be measured are the types of decay, indicated by a decrease in PL after excitation [2]. This technique has been widely used in the analysis of biomolecular structures and the nitrogen-vacancy defects of diamond materials (e.g., [3,4]).

Spinel crystallizes in a cubic crystal system, belonging to the mineral class of "oxides and hydroxides", and possesses the idealized chemical composition of $MgAl_2O_4$, wherein Mg is a divalent cation at the tetrahedrally coordinated site and Al is trivalent and situated at the octahedrally coordinated site. Red spinel is a variant colored by Cr. The PL spectra of spinel are widely used to distinguish natural from flux lab-grown spinel or heated spinel (e.g., [5,6]). However, flux lab-grown and heated spinel still show overlapping Cr PL spectra and require additional chemical testing or gemological observation to separate them.

Chrysoberyl, $BeAl_2O_4$, is an orthorhombic mineral crystallizing in the space group Pbmn [7]. Chrysoberyl and spinel both possess close-packed oxygen structures, wherein chrysoberyl's is hexagonally close-packed (hcp) and spinel's is cubic close-packed (ccp). Chrysoberyl has a lower structural symmetry than spinel because $Be^{2+}$ is a significantly smaller cation than $Al^{3+}$ [8]. Beryllium occupies tetrahedra and Al occupies two types of slightly distorted interstitial octahedral sites: B1 octahedra with Ci symmetry and B2 octahedra with Cs symmetry [9]. These octahedra have different volumes; additionally, B2 has an Al–O distance of 1.936 Å, which is larger than B1, with an Al–O distance of 1.890 Å [10]. Alexandrite is a type of chromium-doped bearing chrysoberyl in which $Cr^{3+}$ is substituted in the $Al^{3+}$ sites.

This study attempted to measure the PL lifetime of spinel and alexandrite. Furthermore, decay profiles were used to characterize their different variants: natural, flux

lab-grown and heated spinel, and natural and lab-grown alexandrite. Time-resolved PL spectroscopy has the potential to be applied as an additional luminescence-testing technique for gemstone identification.

Spinel has the most characteristic photoluminescence spectra ranging from 650 to 750 nm. Gaft et al. assigned luminescence bands at 677, 685, 697, 710, and 718 nm to $Cr^{3+}$ [11]. The most intensive luminescence duplet of the studied sample, which was located at 685 and 687 nm, belonged to the R1 and R2 lines. The R lines were purely electronic and were collectively assigned to the most intense centers. The R lines arose from the state of the $Cr^{3+}$ ion, which corresponded to an ideal short-range order due to the spin-forbidden transition [12,13]. The most intensive peak of natural spinel was located at 685 nm, while those of the flux lab-grown and heated natural spinel were located at 687 nm, showing broader peak widths (Figure 1).

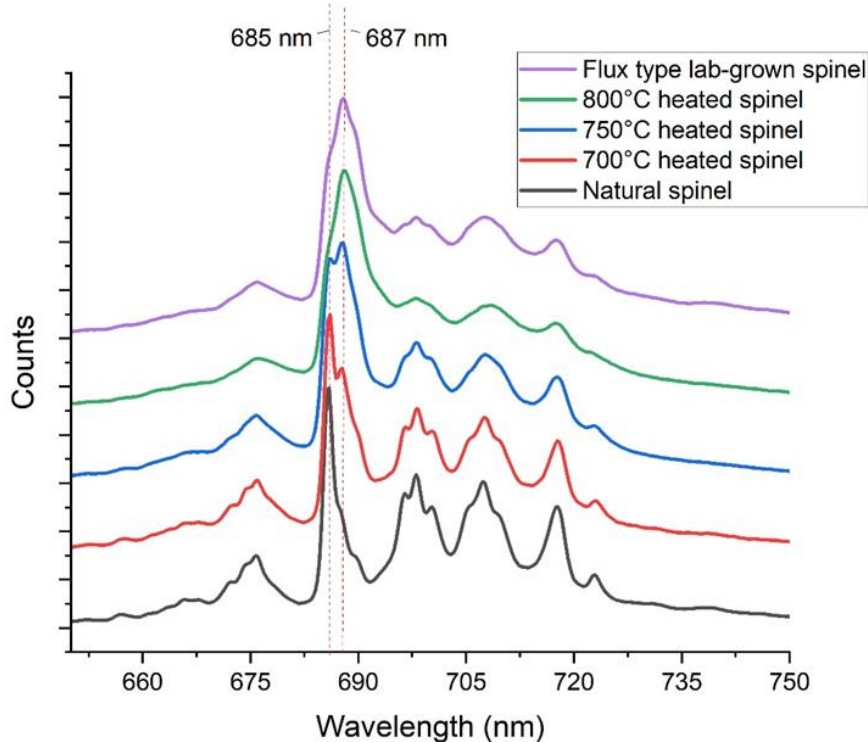

**Figure 1.** Photoluminescence spectra of the natural, heated, and flux lab-grown spinel.

Several powder- or single-crystal diffraction studies on the $MgAl_2O_4$ spinel after quenching from high temperature determined that high temperatures cause the spinel to exhibit order–disorder behavior (e.g., [14–21]). A Mg cation exchanges its site with an Al cation; this formula can be described as $(Mg_{1-x}Al_x)M(Al_{2-x}Mg_x)O_4$, wherein x is between 0 and 1 [15]. Flux lab-grown spinel is crystallized from low-temperature melting at around 1000 °C; therefore, it presents the same degree of disorder in its crystal structure as the heated natural spinel. Since the luminescence properties, radiative transition characteristics, and emissions at selective site excitation depend on the symmetry of the luminescence center environment [11], the $Cr^{3+}$ photoluminescence spectra of the natural heated and lab-grown spinel showed corresponding modifications, which were caused by their respective heating temperatures [12,13,22–28].

There are two types of $Cr^{3+}$ luminescence centers present in alexandrite: R-lines at approximately 679 with 677.3 nm and at 694.4 with 691.7 nm, which are accompanied by N-lines of Cr–Cr pairs located at 644, 650, 653, 667, 669, 678, 680, 690, 694, 702, 707, and 716 nm [11,29,30] (Figure 2). The luminescence band located at 702 nm may be attributed to $V^{2+}$ [11]. The R-lines in chrysoberyl were divided into the Rm (mirror, B2 octahedron) and Ri (inversion, B1 octahedron) lines, which were located at 678 nm (Rm-line) and

690 nm (Ri-line). Numerous experiments have indicated the uneven distribution of cations between the two non-equivalent octahedral sites in chrysoberyl's structure [6,31,32]. The most intensive luminescence duplet at 679 nm was caused by the high $Cr^{3+}$ content at the B2 site with Cs symmetry, where about 70% of the chromium is present [33]. Ferric iron prefers the more symmetrical B1 site, whereas the trivalent chromium cations responsible for the color of alexandrite occupy the B2 site [34].

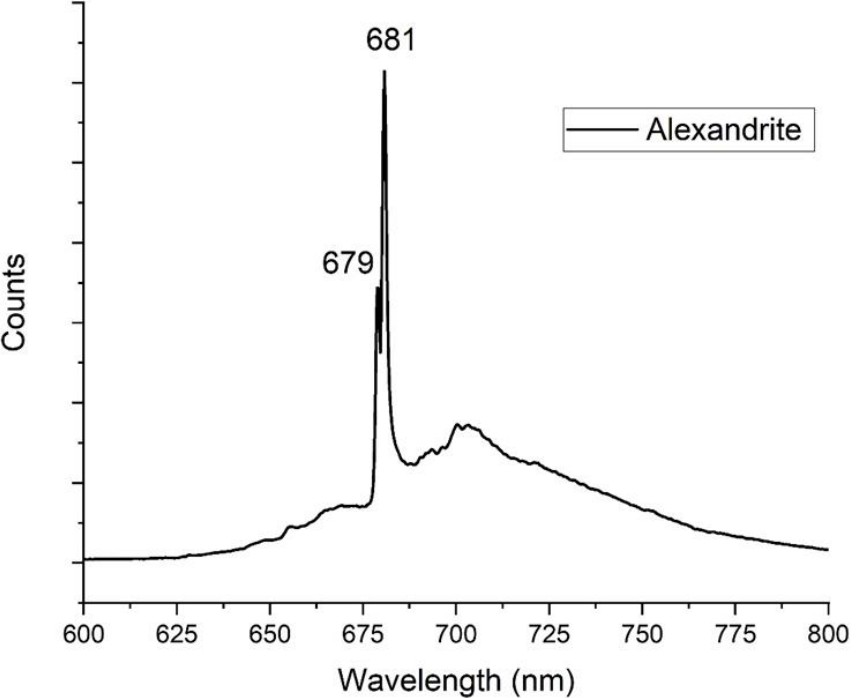

**Figure 2.** Photoluminescence spectra of alexandrite.

The photoluminescence lifetime is the time measured for the excited energy to decay exponentially to N/e (36.8%) of the original state via the loss of energy through fluorescence or non-radiative processes. The fluorescence lifetime is affected by external factors such as temperature, polarity, and the presence of fluorescence quenchers, and also depends on the internal structure of the fluorophore complexity [35]. In this study, all the PL decay was measured of the gem-quality mineral material at room temperature.

## 2. Materials and Methods

In this study, eight gem-quality, faceted, and well-formed octahedral, rough, natural, red-to-pink spinel were selected that originated from Burma and Sri Lanka. To avoid the use of chemically homogenous materials, we deliberately selected spinel from different origins. Additionally, four flux-type lab-grown spinel were selected. Both natural and flux-type lab-grown spinel had the stoichiometry ratio of $MgO:Al_2O_3$ 1:1.

To create heated natural spinel, three rough spinel samples were cut in half and heated in air. The goal of the heat experiments was to generate fluorescence with different ratios between the 685 and 687 nm peaks. Changes in the photoluminescence spectra after heat treatment have previously been observed and attributed to the structural rearrangement during tempering [5,6,22,36]: the peaks at 685 nm and 687 nm changed the relative intensities in the range between 700 and 800 °C; above 800 °C, the R-line peak completely shifted to 687 nm, and there were no further changes in the photoluminescence spectra. Subsequently, the heating temperatures were set at three levels: 700 °C, 750 °C, and 800 °C for 24 h.

Moreover, six natural alexandrite and five lab-grown alexandrite samples with mixed origins were studied. The alexandrite samples were all of a faceted gem quality. Further information on the samples is listed in Table 1.

**Table 1.** List of the selected spinel and alexandrite samples.

| Sample | Type | Shape | Weight (Carats) | Color |
|--------|------|-------|-----------------|-------|
| NatSp1 | Natural spinel | Rough | 0.89 | pink |
| NatSp2 | Natural spinel | Rough | 0.75 | pink |
| NatSp3 | Natural spinel | Rough | 0.92 | pink |
| NatSp4 | Natural spinel | Faceted | 3.56 | Purple red |
| NatSp5 | Natural spinel | Faceted | 5.12 | Red |
| NatSp6 | Natural spinel | Rough | 1.18 | Pink |
| NatSp7 | Natural spinel | Rough | 1.32 | Pink |
| NatSp8 | Natural spinel | Rough | 1.15 | Pink |
| FLGSp1 | Lab-grown spinel | Faceted | 2.14 | Red |
| FLGSp2 | Lab-grown spinel | Polished Slab | 0.53 | Red |
| FLGSp3 | Lab-grown spinel | Polished Slab | 0.56 | Red |
| FLGSp4 | Lab-grown spinel | Polished Slab | 0.76 | Red |
| NatAx1 | Natural alexandrite | Faceted | 1.23 | Dark green to purple |
| NatAx2 | Natural alexandrite | Faceted | 1.43 | Dark green to purple |
| NatAx3 | Natural alexandrite | Faceted | 1.35 | Dark green to purple |
| NatAx4 | Natural alexandrite | Faceted | 1.56 | Dark green to purple |
| NatAx5 | Natural alexandrite | Faceted | 1.11 | Dark green to purple |
| NatAx6 | Natural alexandrite | Faceted | 1.89 | Dark green to purple |
| LGAx1 | Lab-grown alexandrite | Faceted | 2.01 | Green to red purple |
| LGAx2 | Lab-grown alexandrite | Faceted | 1.98 | Green to red |
| LGAx3 | Lab-grown alexandrite | Faceted | 2.56 | Green to purple |
| LGAx4 | Lab-grown alexandrite | Faceted | 2.98 | Green to red |
| LGAx5 | Lab-grown alexandrite | Faceted | 1.34 | Green to purple |

In this study, we selected a 405 nm laser (FC-D-405-50 mW, CNI) as the excitation source. The custom made 405 nm spectroscopy probe (SPC-R405, Spectra Solution) guided the laser and utilized optics with a 9 mm working distance to create an approximately 100 um focal spot to excite and collect the PL signal from the sample. Three optical filters were selected to isolate the excitation from the emission signal including a laser clean-up filter (LL01-405, Semrock) that precisely defined the excitation wavelength, a dichroic beam splitter (FF409-Di03, Semrock) to reflect the laser to the sample while passing the PL signal, and a long pass filter (LP02-407RU-25) to provide additional cut-off to the excitation. Finally, a compact miniature spectrometer (Avaspec-mini, Avantes) with 1.2 nm spectral resolution and 400 to 900 nm sensing range was used to collect the PL signal. Two 100 um optical fibers were used to guide the laser to the spectroscopy probe and to relay the spectra to the spectrometer. The spectroscopy probe was pre-aligned to the center of a black color anodized metal aperture sample stage with a 0.7 mm diameter.

In order to analyze the PL decay, a series of PL spectra using a millisecond integration time were continuously collected. The laser was shut down shortly after the beginning of the spectra collection to initiate the PL decay. The collected spectra were stored to the random-access memory (RAM) of the spectrometer to accurately label the time stamp on each spectrum with a minimized sampling interval. The sampling interval between each spectrum was observed between 0.6 to 1.6 ms, mainly caused by the electronic spectrometer. The spectra collection and the time stamp labeling was automatically performed by the Avasoft 8.8 software provided by the manufacturer of the spectrometer. This PL lifetime spectroscopy can serve as a low-cost (<$10,000) alternative setup compared to other sophisticated time-resolved techniques such as time-correlated single photon counting (TCSPC) and time-gated spectroscopy.

## 3. Results

### 3.1. Photoluminescence Decay Spectra and Duration of Decay

#### 3.1.1. Orientational Homogeneity

First, a series of orientational homogeneity tests were conducted. Notably it is important to determine whether the results of the duration of decay and decay profile were

repeatable for each sample regardless of their orientation as well as show that the 405 PL device produced sufficiently repeatable results. The natural spinel sample NatSp3 is an octahedral-form rough crystal. With the laser vertically oriented with respect to each crystal facet, the decay spectra were collected six times.

The R-line band at 685 nm was the most persistent and was the last peak to disappear. The intensities of the entire spectral band series decreased at the same speed, and the intensity ratios between each band remained unchanged until they declined under the background (Figure 3). The decay curve of the 685 nm band intensity was used to represent each decay event. All of the curves were normalized to 100. The six decay curves of each measurement illustrated in Figure 4 were almost identical. The repeat tests show that the type of the photoluminescence decay was independent of orientation, and the decay spectra, the duration of decay, and the decay curve's profile were all reproducible by the 405 PL device. The duration of decay values of each measurement with respect to sample NatSp3 are depicted in Figure 4; they resulted in an average PL lifetime of 21 ms with a standard error of 0.8 ms.

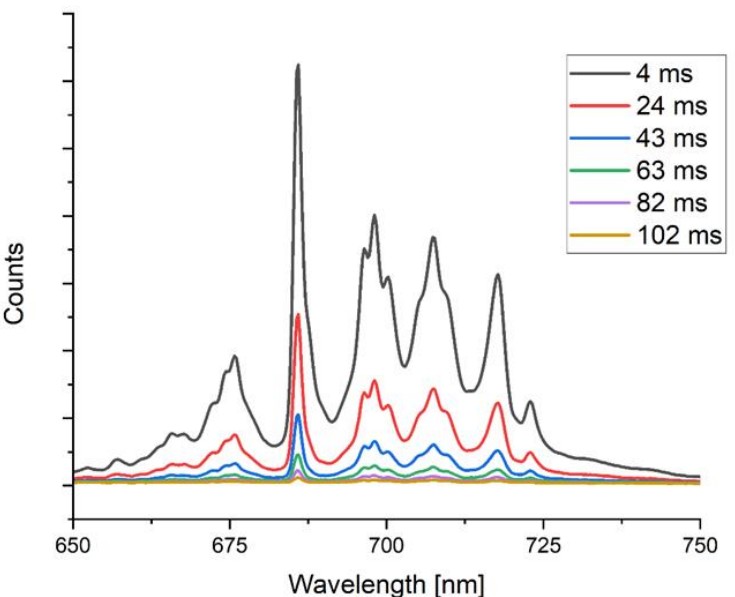

**Figure 3.** Time-resolved fluorescence decay spectra of natural spinel.

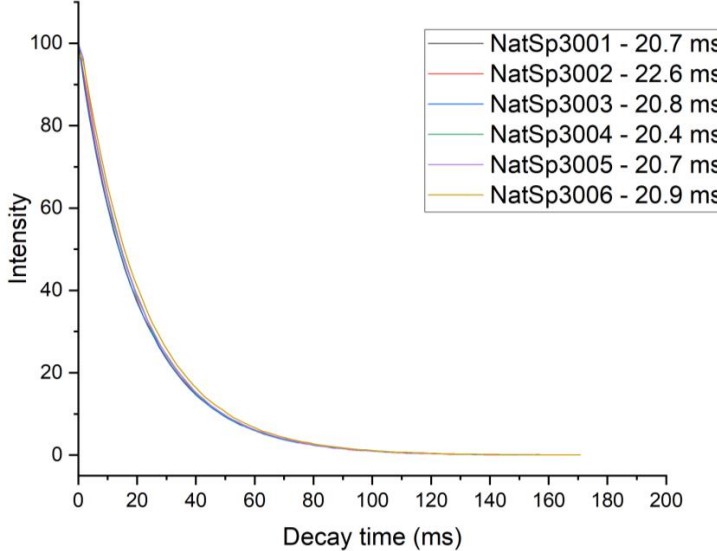

**Figure 4.** Six photoluminescence decay curves at 685 nm of the rough spinel NatSp3.

Although the 405 PL device was not specifically built for time-resolved testing, in our opinion, it is sufficient for this preliminary study because the luminescence of spinel lasts long enough. Accordingly, a 3%–4% standard error is acceptable.

### 3.1.2. Comparison Natural Spinel, Heated Spinel, and Flux Lab-Grown Spinel

Heated natural spinel and flux lab-grown spinel are different from unheated natural spinel and are characterized by a dominant R-line band at 687 nm, with a broader peak width, which revealed an overlapping of both R-line bands at 685 and 687 nm (Figure 1). The same interpretation was made by [6], which was quantified via Gaussian deconvolution. Interestingly, the luminescence decay spectra proved the existence of double peaks. The 687 nm band showed a more rapidly decreasing speed than the 685 nm band. Accordingly, the decay spectra of the flux lab-grown and heated natural spinel began with dominant bands at 687 nm, but as the luminescence decayed, the dominant peak position shifted to around 685 nm with a simultaneous decrease in the peak width. In the middle of the decay process, double peaks could also be observed (Figure 5).

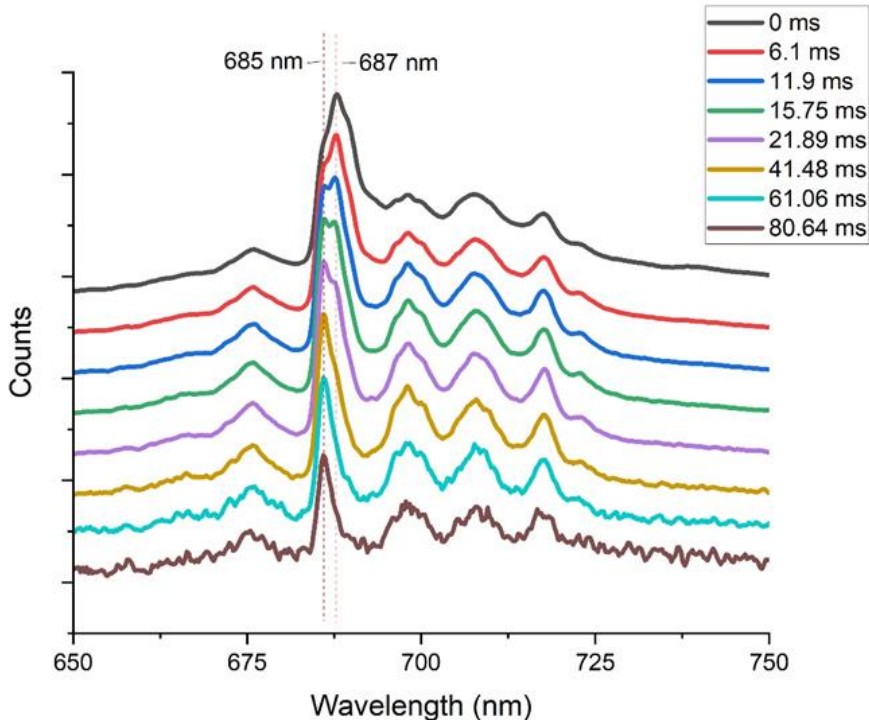

**Figure 5.** Time-resolved fluorescence decay spectra of the flux lab grown and heated spinel.

Regarding the PL lifetime, there was separation in the range of decay times between the natural spinel, heated spinel, and flux lab-grown spinel (see Table 2). The eight natural, unheated spinel samples had a PL lifetime between 12.8 ms and 23.4 ms, with an average of 18.7 ms and a standard error of 3.1 ms. The three heated natural spinel samples all showed decreased durations of decay compared to their unheated counterparts, with an average duration of decay of 14 ms; nevertheless, these values were still within the range of natural spinel's duration of decay. This also revealed that the crystal-structure-disordering process caused a reduction in the fluorescence-related duration of decay under the same chemical environment. The four flux lab-grown spinel samples had an average duration of decay of $8.87 \pm 2.41$ ms. Aizawa et al. showed very a similar lifetime of lab-grown spinel of 10 ms [37]. In comparison, the PL lifetime values of the natural unheated and heated spinel were generally longer than the flux lab-grown spinel. Due to the overlap between the heated natural spinel and flux lab-grown spinel using only single fluorescence spectra, this result is meaningful (Figure 6).

**Table 2.** Summary of the decay duration and exponential parameter half-life of the spinel and alexandrite samples. Half-life A1 ($t_{1/2}$) and half-life A2 ($t_{1/2}$) corresponded to the one-phase and two-phase exponential function fitting parameters, respectively. A1 and A2 are the coefficients to the exponential function.

| Type | Sample | Lifetime (ms) | Exponential Fitting Parameters | | | | | |
| | | | Half-Life A1 ($t_{1/2}$) (ms) | Standard Error | Half-life A2 ($t_{1/2}$) (ms) | Standard Error | A1 | A2 |
|---|---|---|---|---|---|---|---|---|
| Natural spinel | NatSp1 | **17.5** | 11.8 | 0.11 | | | | |
| | NatSp2 | **18.7** | 13.2 | 0.07 | | | | |
| | NatSp3 | **20.5** | 14.4 | 0.05 | | | | |
| | NatSp4 | **23.4** | 15.2 | 0.13 | | | | |
| | NatSp5 | **12.8** | 8.7 | 0.14 | | | | |
| | NatSp6 | **19.6** | 13.5 | 0.06 | | | | |
| | NatSp7 | **20.3** | 14.3 | 0.05 | | | | |
| | NatSp8 | **16.9** | 11.8 | 0.07 | | | | |
| | Average | **18.7** | 12.9 | | | | | |
| | Standard error | **3.1** | 2.1 | | | | | |
| Natural heated spinel | NatSp6 800 | **11.0** | 14.8 | 1.87 | 5.57 | 0.33 | 0.28 | 0.73 |
| | NatSp7 750 | **15.5** | 15.2 | 1.17 | 6.4 | 0.65 | 0.54 | 0.47 |
| | NatSp8 700 | **15.4** | 12.8 | 0.08 | 4.34 | 0.13 | 0.78 | 0.21 |
| | Average | **14.0** | 14.3 | | | | | |
| | Standard error | **2.6** | 1.2 | | | | | |
| Flux lab-grown spinel | FLGSp1 | **7.2** | 8.4 | 0.12 | 2.22 | 0.06 | 0.52 | 0.47 |
| | FLGSp2 | **12.3** | 14.5 | 1.61 | 5.89 | 0.41 | 0.36 | 0.65 |
| | FLGSp3 | **8.9** | 10.1 | 0.19 | 3.1 | 0.08 | 0.52 | 0.48 |
| | FLGSp4 | **7.1** | 42.1 | 160.9 | 4.35 | 0.34 | 0.05 | 1.02 |
| | Average | **8.9** | 18.8 | | | | | |
| | Standard error | **2.4** | 17.4 | | | | | |
| Natural alexandrite | NatAx1 | **30.5** | 0.5 | 0 | 19.73 | 0.5 | | |
| | NatAx2 | **25.7** | 0.7 | 0.03 | 17.56 | 1.08 | | |
| | NatAx3 | **28.7** | 0.2 | 0.01 | 20.75 | 0.89 | | |
| | NatAx4 | **43.9** | 0.5 | 0.02 | 24.37 | 0.22 | | |
| | NatAx5 | **27.5** | 0.4 | 0.04 | 17.41 | 0.38 | | |
| | NatAx6 | **26.3** | 2.1 | 1.63 | 14.03 | 0.72 | | |
| | Average | **30.4** | 0.7 | | 18.97 | | | |
| | Standard error | **6.8** | 0.7 | | 3.51 | | | |
| Lab-grown alexandrite | LGAx1 | **49.3** | 0.9 | 0.01 | 34.67 | 1.85 | | |
| | LGAx2 | **31.8** | 0.8 | 0.03 | 16.79 | 0.41 | | |
| | LGAx3 | **32.1** | 4.2 | 8.05 | 20.05 | 0.83 | | |
| | LGAx4 | **44.3** | 2.7 | 2.8 | 31.03 | 0.53 | | |
| | LGAx5 | **52.8** | 0.5 | 0.03 | 31.14 | 0.49 | | |
| | Average | **42.1** | 1.8 | | 26.74 | | | |
| | Standard error | **9.7** | 1.6 | | 7.82 | | | |

### 3.1.3. Alexandrite's Decay Spectra and Duration of Decay

The 405 nm laser-excited decay spectra enabled us to detect two different $Cr^{3+}$ centers, which presented as two distinguishable decay processes. The eleven alexandrite samples showed consistent properties. The first decay step occurred at the dominant duplet R-lines at 681 and 679 in the excitation state (Figure 2), which were caused by about 70% of the $Cr^{3+}$ content at the B2 site [33]. These bands diminished into the background rapidly within the first 5 ms. After the first decay step, a series of $Cr^{3+}$ peaks in the B1 site bands appeared, with major bands at 690 and 696 nm (Figure 7). The decay process of the B1 site bands lasted much longer, contributing to the majority of the duration of decay of the whole process. The 696 nm band disappeared, thus marking it as the last emission band. Walling et al. also determined a two-step photoluminescence decay process of alexandrite crystal, a 2.3 ms

lifetime of the mirror site emission, and the inversion-site emission had a longer lifetime of 48 ms.

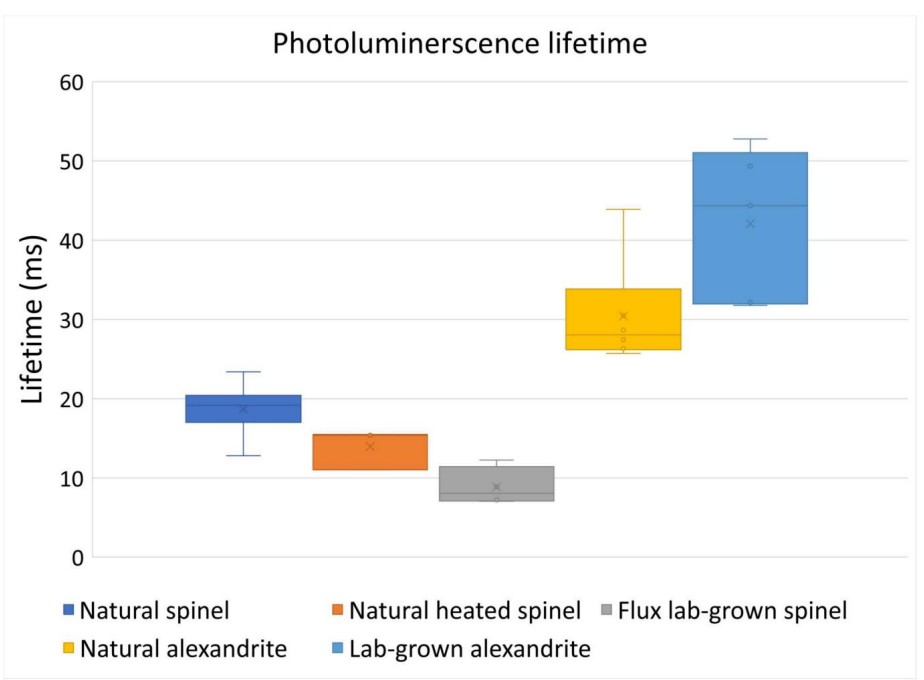

**Figure 6.** PL lifetime of the analyzed spinel and alexandrite samples.

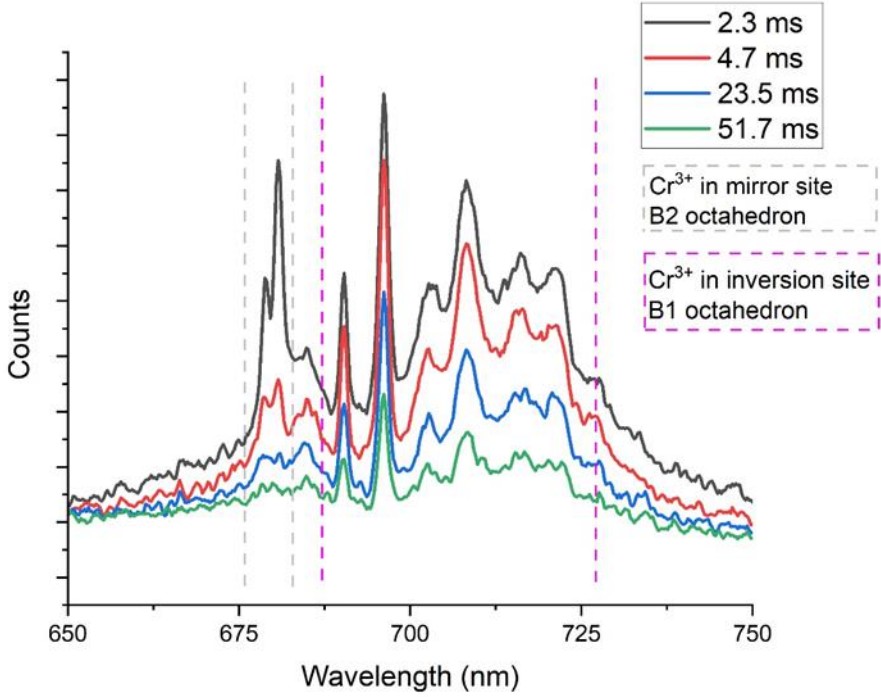

**Figure 7.** Time-resolved fluorescence decay spectra of alexandrite.

Due to the excessively short decay time of the B2 site, there were no more than five measurements that could be collected on the first stage of decay. The PL lifetime is represent by the B1 site decay. The lifetime of the six natural alexandrite samples was between 25.7 and 43.9 ms with an average of 30.43 ms. In comparison, the lifetimes of the lab-grown alexandrite samples were higher with an overlap with the latter values between 31.8 and 52.8 ms. The PL lifetime of the natural and lab-grown alexandrite showed a significant separation between the two groups with a limited overlap zone (Figure 6).

*3.2. Decay Curve Fitting with Exponential Function*

To quantify the curve profile, an exponential decay function was applied to fit the curves. An exponential decay curve fits the following equation:

$$y = e^{-t/\tau} \tag{1}$$

where $t$ is the time and $\tau$ is the decay constant. The half-life of the decay is related to the decay constants in the following way:

$$t_{1/2} = In(2)\tau \tag{2}$$

where $t_{1/2}$ is the half-life. The half-life of a decay curve quantifies the decay speed of each measurement.

Two-phase exponential decay functions are applied when a one-phase exponential decay function cannot fit the curve's shape.

The formula for a one-phase exponential decay function is as follows:

$$y = A1 \times e^{-\frac{t}{\tau}} + y0 \tag{3}$$

The formula for a two-phase exponential decay function is as follows:

$$y = A1 \times e^{-\frac{t}{\tau 1}} + A2 \times e^{-\frac{t}{\tau 2}} + y0 \tag{4}$$

### 3.2.1. Spinel Decay Curve Fitting

The decay curves of all the tested spinel samples are presented in Figure 8 and Table S1. Not only was the PL lifetime, but the decay curve shapes were also different between the natural spinel and flux lab-grown spinel, and the curves of the natural spinel displayed a more obvious gentle slope than lab-grown spinel. Heated spinel samples NatSp6, 7, and 8 illustrated a visible increase in the slope in comparison with their unheated part, located between the natural and flux lab-grown spinel curves.

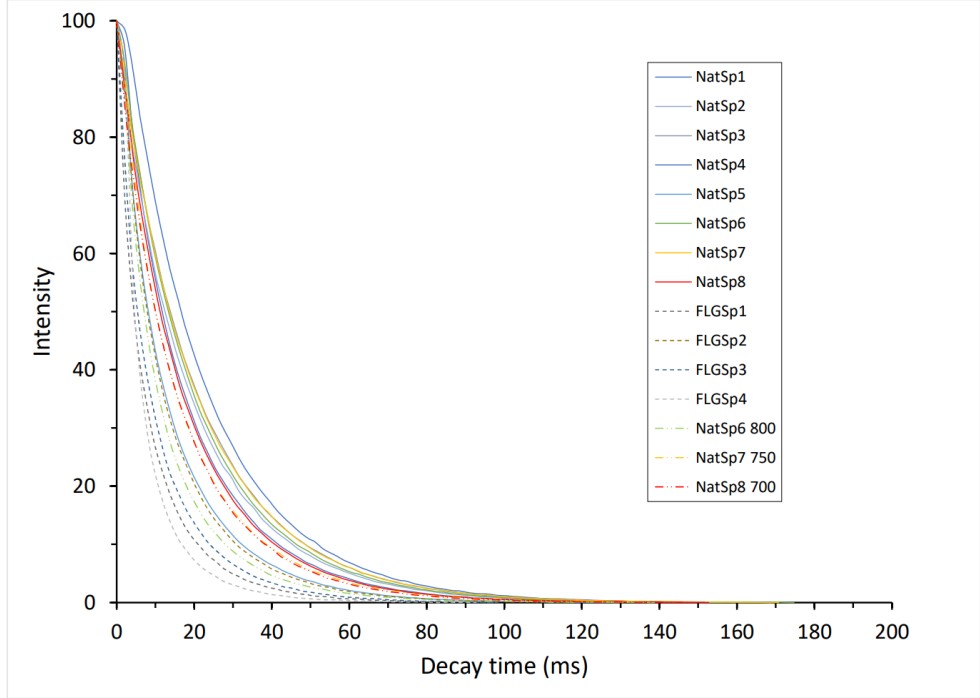

**Figure 8.** The photoluminescence decay curves at 685 nm of spinel: natural spinel versus the heated and flux lab-grown spinel samples.

The decay half-lives of the fitting parameters ($t_{1/2}$) together with their standard errors are listed in Table 2. The fit of the natural spinel's decay curves with respect to the one-phase exponential function was suitable for all samples. However, the flux lab-grown and heated spinel all needed to be fit using the two-phase exponential function and are listed under half-life $t_{1/2}$ A1 and half-life $t_{1/2}$ A2 in Table 2. This result coincides with the observation that the dominant fluorescence band at 687 nm band showed a higher decay speed in the two spinel groups.

The fitting results revealed that only one decay path was present in natural spinel and followed the exponential decay process, while in the lab-grown and heated spinel crystal structures, two decay paths existed, and the two decay paths were differentiated by their decay speeds. The site exchange between a Mg cation and an Al cation created the second decay path. $Mg^{2+}$ at the octahedrally coordinated and $Al^{3+}$ at the tetrahedrally coordinated sites will create two inequivalent lattice cells, whose presence is more conducive to photon transmission. These paths can be termed as "normal" and "inverse" paths, respectively. Switching between the $Mg^{2+}$ cation and $Al^{3+}$ cation increases the proportion of the "inverse" decay path.

The coefficients A1 and A2 in the two-phase exponential fitting function refer to the proportions of "normal" decay and "inverse" decay, respectively. A1/A2 decreased correspondingly: 78/21, 54/47, and 28/73 via the three spinel samples heated at 700 °C, 750 °C, and 800 °C. This corresponds to spinel's order–disorder behavior theory, in which, with respect to $(Mg_{1-x}Al_x)M(Al_{2-x}Mg_x)O_4$, x increases through heating [15]. However, this does not mean that the A1/A2 ratio can be directly transferred to x, but proves that the decay curve fitting results are valid and reflect the order–disorder characteristics of the spinel crystals.

Through exponential fitting, it was concluded that the natural, unheated spinel had an average half-life ($t_{1/2}$) of 13.6 ± 1.9 ms. The two-phase exponential fitting of the heated spinel data split two decay processes: a slow decay with a half-life from 12.8 to 15.1 ms and a quick decay with a half-life from 4.3 to 6.4 ms. The slow decay's parameters were very close to the unheated samples, and it corresponded to a "normal" decay path. The quicker decay corresponded to "inverse" decay.

The exponential fitting of the two components of the flux lab-grown samples also resulted in one "slow" and one "quick" decay. However, their half-life values showed a different combination compared to the heated natural sample.

In summary, natural spinel is characterized by one-phase exponential decay, whereas heated and flux lab-grown spinel follow two-phase exponential decay.

### 3.2.2. Alexandrite Decay Curve Fitting

The 696 nm band decay curves of the alexandrite samples are displayed in Figure 9 and Table S2. As a result of the two-component decay process with greatly different decay speeds between the B1 and B2 sites, both decay curves were entirely bent, presenting an obvious angular appearance. The decay curves of the natural and lab-grown alexandrite can be divided by their visibility at the angular area (see in Figure 9), while natural alexandrite started their second step by 1%–4% intensity, the lab-grown alexandrite curves displayed a higher bending point between 5 and 10% intensity. This indicates that in the lab-grown material, there were proportionally significantly more Cr sites at the B1 site than in the natural material structure.

The whole decay curve can be visually split up into a two-component exponential decay process. This began with short lifetime B2 site decay followed by long lifetime B1 site decay. The B2 site decay fitting parameters are listed under half-life A1, which resulted in a large standard error. We note that due to the minimal sampling interval of our system, the photoluminescence lifetimes shorter than 5 ms may not have been confidently measured. Based on our measurement, the B2 site decay time was approximately 2 ms.

Alternatively, the characterization of the natural and lab-grown alexandrite decay curves focused on fitting the B1 site's decay process. The use of a one-phase exponential

decay function was suitable for determining the B1 site's decay profile. The fitting parameters' decay half-life ($t_{1/2}$), together with their standard errors, are listed under half-life A2 in Table 2.

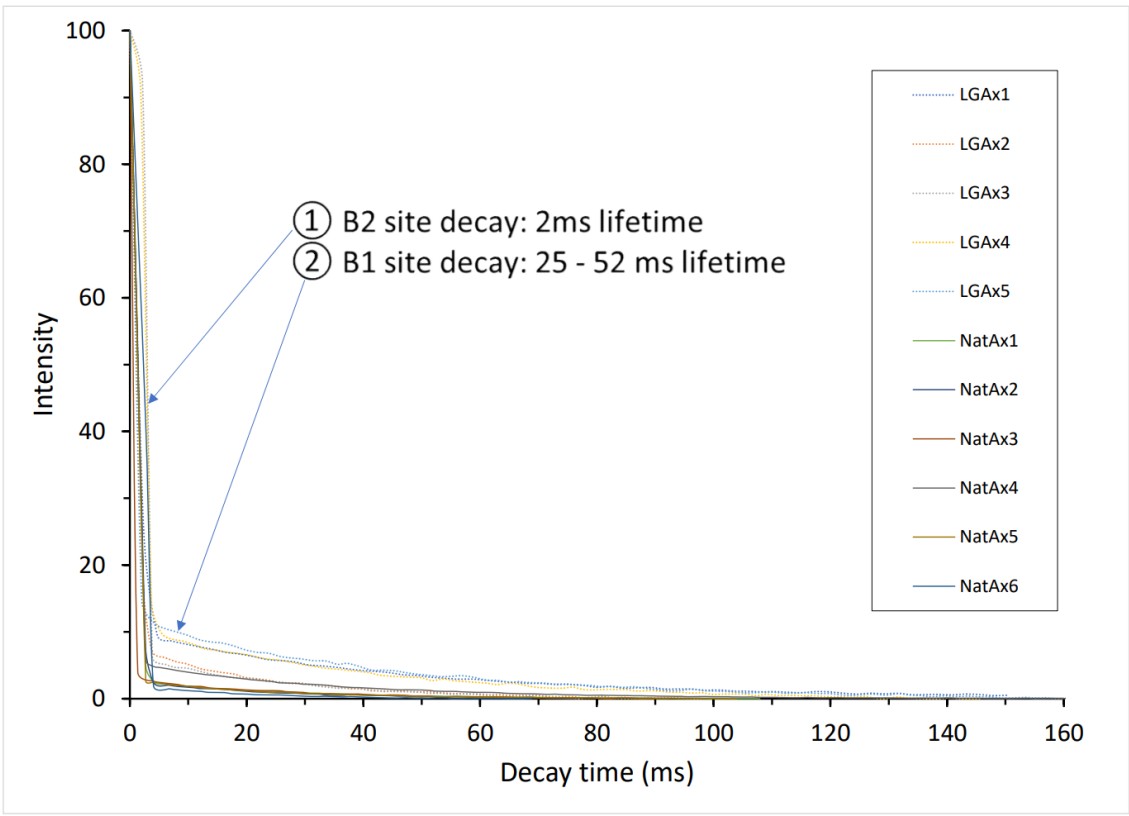

**Figure 9.** Photoluminescence decay curves at 696 nm of the natural and lab-grown alexandrite samples. Indication of the two decay processes that are related to $Cr^{3+}$ in the mirror site (B2) and $Cr^{3+}$ in the inverse site (B1). B1 site decays begin with a different intensity of individual samples.

The evaluation of the half-life ($t_{1/2}$) showed the same resolution between the natural and lab-grown alexandrite compared to the PL lifetime. The decay half-life values of the natural alexandrite's B1 site were between 14 and 24 ms, with an average of 18.9 ms, whereas those of the lab-grown alexandrite samples were between 16.8 and 34.7 ms, with an average 26.7 ms. On average, the lab-grown alexandrite's decay speed was lower than the natural one. Among the five lab-grown samples, three resulted in half-lives over 30 ms.

In contrast to spinel, the natural and lab-grown alexandrite did not show any differences in any emission, Raman, or fluorescence spectra. Ollier et al. (2015) [38] pointed out that the $Fe^{3+}$ ions were efficiently substituted in the mirror site, and had a strong impact on the $Cr^{3+}$-related lifetime of the mirror site. This explains the differences in the half-life parameter between the natural and lab-grown alexandrite observed herein. Natural alexandrite from different origins contains between 600 and 10,000 ppm of iron (e.g., [39–42]). Generally, lab-grown gem material contains less iron in comparison to its natural varieties. In some lab-grown alexandrite, impurities of iron can be under the detection limit of ICPMS or EDXRF. In our case, we were able to use the laser-induced time-resolved photoluminescence decay profile to distinguish the low-iron-content lab-grown alexandrite from the high-iron content natural alexandrite. Furthermore, the PL decay curve indicated significant higher $Cr^{3+}$ contents in the mirror site of the lab-grown alexandrite compared to the natural alexandrite, which can be used as a second characteristic criterion. Plotting the PL lifetime versus the intensity value at a 7.05 ms decay time where the junction point between the B1 and B2 site decay processes in Figure 10 significantly enhanced the separation between the two groups.

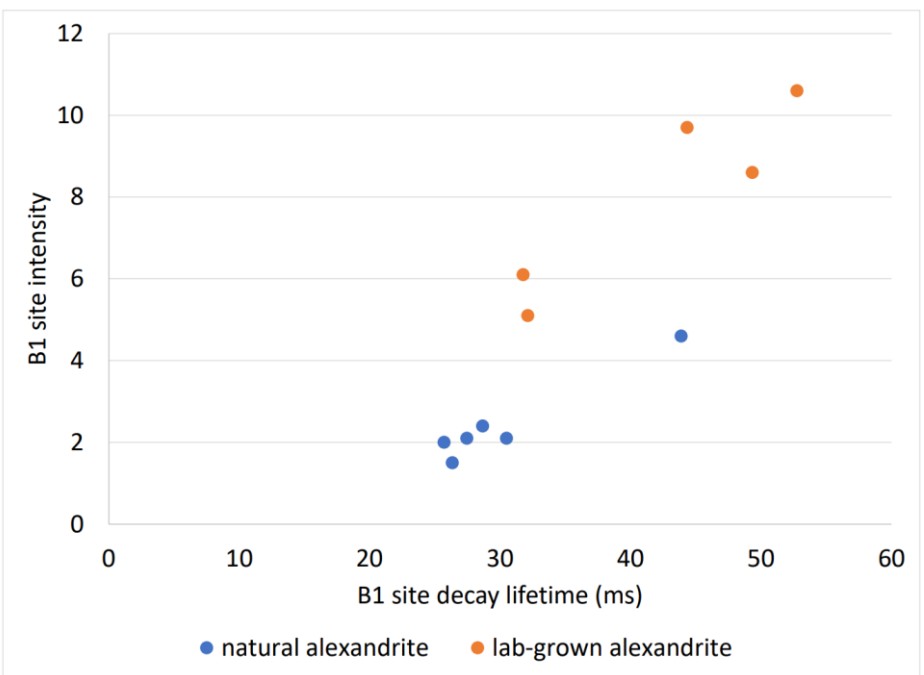

**Figure 10.** Plotting the B1 site decay lifetime with the B1 site starting intensity. This resulted in a better separation between the natural and lab-grown alexandrite.

## 4. Discussion

Spinel and alexandrite both exhibited a long photoluminescence lifetime up to 23.4 ms and 52.8 ms. Another Cr-doped gemstone species showing a long fluorescence lifetime is ruby. Similar fluorescence decay experiments demonstrated ruby's fluorescence lifetime to be about 3.5 ms [43], which is 1/5 to 1/10 that of spinel and alexandrite. There are crystallographic commonalities among ruby, spinel, and chrysoberyl: (1) they all possess a close-packed oxygen structure; and (2) $Cr^{3+}$ is substituted for $Al^{3+}$ in the octahedral sites. The major difference between their structural complexities is that two thirds of ruby's octahedral sites are occupied while an eighth of the tetrahedral holes and half of the octahedral holes are occupied in spinel and alexandrite crystals [7]. Structural complexity governs the number of decay paths available to electrons and the duration for which electrons remain in an excited state. The case of heated spinel demonstrated that the modification of the disorder of the lattice on different levels dramatically changes the decay profiles. This property provides a potential application for investigations into gemstone treatments.

Another aspect influencing a gemstone's fluorescence lifetime is chemistry, specifically, the aluminum substitution elements of chromium versus the iron content, for which the duration of decay of alexandrite is partially related to its iron concentration, as has been demonstrated in another study [22]. Iron concentration is a very important element in the comparison of natural and lab-grown gemstone variants, but also gem-materials from different origins. A typical example is the use of iron to distinguish marble-type rubies from metamorphic original rubies (e.g., [44–47]). The correlation between chemical patterns and luminescence lifetime profiles requires further investigation with more statistical data.

This experimental, primary study based on the 405 nm PL device is highly instructive. Through its completion, we have obtained a better understanding of the capabilities and applications of laser-introduced time-resolved photoluminescence in terms of gemstone identification, treatment determination, and, potentially, origin determination. Further investigations (e.g., using nanosecond measurements and more sophisticated instrumentation) will increase the availability of methods for distinguishing different groups of gemstones. Unlike traditional gemstone-testing methods, the quantitative time-resolved luminescence method is capable of assessing large quantities of data. Ideally, together

with the automation of data acquisition and analysis software, this method can fulfill the requirements of gemstone-testing analyses.

## 5. Conclusions

Using 405 nm laser-induced, time-resolved photoluminescence decay spectroscopy to study dozens of reference samples of natural and lab-grown spinel and alexandrite as well as heated spinel, we determined that both spinel and alexandrite possess long photoluminescence lifetime properties of up to 23.4 ms and 52.8 ms. Detailed investigation into the samples' decay spectra profiles clearly separated the natural, heated natural, and flux lab-grown spinel by their duration of decay and exponentially fitted half-lives ($t_{1/2}$). Natural and lab-grown alexandrite also possess different half-life ($t_{1/2}$) patterns due to their diverse impurity chemical components.

**Supplementary Materials:** The following supporting information can be downloaded at: https://www.mdpi.com/article/10.3390/min13030419/s1, Table S1: Decay curves of the analyzed spinel samples; Table S2: Decay curves of the analyzed alexandrite samples.

**Author Contributions:** Conceptualization, W.X.; Methodology, W.X. and T.-H.T.; Software, W.X. and T.-H.T.; Formal analysis, W.X.; Investigation, W.X.; Resources, W.X., T.-H.T. and A.P.; Data curation, W.X. and T.-H.T.; Writing—original draft preparation, W.X.; Writing—review and editing, W.X. and T.-H.T.; Project administration, W.X. and A.P. All authors have read and agreed to the published version of the manuscript.

**Funding:** This research received no external funding.

**Data Availability Statement:** All data generated or used during the study appear in the submitted article.

**Conflicts of Interest:** The authors declare no conflict of interest.

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
