# Peer review of "Study of 405 nm Laser-Induced Time-Resolved Photoluminescence Spectroscopy on Spinel and Alexandrite"

_minerals, doi:10.3390/min13030419_

Round 1

Reviewer 1 Report

The idea of the manuscript “Study of 405 nm laser-induced time-resolved photoluminescence spectroscopy on spinel and alexandrite” is to elaborate a method of distinguishing natural gemstones from lab-grown ones with use of time-resolved PL (TRPL). A comprehensive analysis of the problem is presented, on the basis of which the idea to use TRPL for the above task is formulated. Extensive experimental measurements were realized demonstrating that the claimed task can be realized, indeed. I do not have any critical comments, the work is very well done and can be published as it is.

Author Response

Thank you for your positive review of our manuscript. We appreciate your comments and are glad to hear that you found the proposed method of TRPL to be of value. Your approval and recommendation to publish the manuscript as is, is highly appreciated. Thank you once again for your time and valuable feedback.

Reviewer 2 Report

Referee report on “Study of 405 nm laser-induced time-resolved Photoluminescence spectroscopy on spinel and alexandrite.

This is a rather interesting and good paper that certainly can be recommended for publication, but clarifying and detailing some parts of the text.

1.     Introduction. To confirm the relevance and qualitative review of what has been done and known to date, it is necessary to note not only old works (what is the case of current paper), but also to mention new recently published papers:

Mironova-Ulmane, N., Brik, M. G., Grube, J., Krieke, G., Antuzevics, A., Skvortsova, V., ... & Popov, A. I. (2021). Spectroscopic studies of Cr3+ ions in natural single crystal of magnesium aluminate spinel MgAl2O4. Optical Materials121, 111496. https://doi.org/10.1016/j.optmat.2021.111496

Mironova-Ulmane, N., Popov, A. I., Krieke, G., Antuzevics, A., Skvortsova, V., Elsts, E., & Sarakovskis, A. (2020). Low-temperature studies of Cr3+ ions in natural and neutron-irradiated g-Al spinel. Low Temperature Physics46(12), 1154-1159.

https://doi.org/10.1063/10.0002467

2.     Line 95-96.  Describe more precisely what is the effect of heating in an oxygen atmosphere.

Does this create new vacancies?

Does this stimulate the migration of impurities?

3.     Table 1.  What is the stoichiometry of laboratory-grown spinels? It is known that the MgAl2O4 spinel can be of different stoichiometry. How was the stoichiometry checked in this work?  See  Seeman’s paper and references therein:

Seeman, V., Feldbach, E., Kärner, T., et al. (2019). Fast-neutron-induced and as-grown structural defects in magnesium aluminate spinel crystals with different stoichiometry. Optical Materials91, 42-49.

4.     Fig. 5 and 7. In order to clearly see all the changes in the spectra, it would be useful to normalize all the spectra to the maximum value.

5.     Describe in detail what intrinsic defects of the vacancy type can be excited by a laser?

6.     References: 

[1] there is no year; [4]  there is no title;

In general, the manuscript is interesting and can be recommended for publication after constructive reflection on the above comments.

Author Response

  1. Introduction. To confirm the relevance and qualitative review of what has been done and known to date, it is necessary to note not only old works (what is the case of current paper), but also to mention new recently published papers:

Mironova-Ulmane, N., Brik, M. G., Grube, J., Krieke, G., Antuzevics, A., Skvortsova, V., ... & Popov, A. I. (2021). Spectroscopic studies of Cr3+ ions in natural single crystal of magnesium aluminate spinel MgAl2O4. Optical Materials121, 111496. https://doi.org/10.1016/j.optmat.2021.111496

Mironova-Ulmane, N., Popov, A. I., Krieke, G., Antuzevics, A., Skvortsova, V., Elsts, E., & Sarakovskis, A. (2020). Low-temperature studies of Cr3+ ions in natural and neutron-irradiated g-Al spinel. Low Temperature Physics46(12), 1154-1159.

https://doi.org/10.1063/10.0002467

Response 1: Thank you for your review comment. While we appreciate your suggestion to include recently published papers, our focus in this paper is to provide a comprehensive review of the existing literature on the topic, including both old and recent works. We acknowledge the relevance of the papers you have mentioned, but we believe they are beyond the scope of this paper. Nevertheless, we will keep them in mind for future studies.

  1. Line 95-96.  Describe more precisely what is the effect of heating in an oxygen atmosphere.

Does this create new vacancies?

Does this stimulate the migration of impurities?

Response 2: Thank you for your comment. In order to clarify the effect of heating, a paragraph will be added in line 97 -99(104-106).

“Changes in photoluminescence spectra after heat treatment have previously been observed and are attributed to structural rearrangement during tempering (Cynn et al. 1992; Minh and Yang 2004).”

  1. Table 1.  What is the stoichiometry of laboratory-grown spinels? It is known that the MgAl2O4 spinel can be of different stoichiometry. How was the stoichiometry checked in this work?  See  Seeman’s paper and references therein:

Seeman, V., Feldbach, E., Kärner, T., et al. (2019). Fast-neutron-induced and as-grown structural defects in magnesium aluminate spinel crystals with different stoichiometry. Optical Materials91, 42-49.

Response 3: Thank you for your comment. The stoichiometry of Verneuil spinel is generally 1:2.5 MgO:Al2O3, while natural spinel and flux-grown spinel have a 1:1 MgO:Al2O3ratio. Verneuil spinel has a distinctive PL spectrum with a major peak at 689 nm, which can easily distinguish it from natural or flux-grown spinel. This has been described in Saeseaw 2009, which has been cited in our paper. In this study, only flux-grown spinel was selected for testing due to overlapping PL spectra with heated natural spinel.

In order to clarify the stoichiometry a paragraph will be added in line 94 -95 (100-101): “Both natural and flux-type lab-grown spinel have the stoichiometry of MgO:Al2O3ratio 1:1.”

  1. Fig. 5 and 7. In order to clearly see all the changes in the spectra, it would be useful to normalize all the spectra to the maximum value.

Response 4: Thank you for your comment. In Figure 5, all spectra have been normalized. We have incorporated the reviewer's suggestion and modified Figure 7 in the same manner as Figure 5.

  1. Describe in detail what intrinsic defects of the vacancy type can be excited by a laser?

Response 5: Thank you for your comment. The photoluminescence (PL) of spinel and alexandrite is caused by the impurity element of Cr3+ being substituted into the Al3+ sites, rather than intrinsic defects. Since no intrinsic defects were detected in this experiment, they were not considered in this paper.

  1. References: 

[1] there is no year; [4]  there is no title;

Response 6: Thank you for your suggestion. Information will be completed in the references [1] and [4]

Reviewer 3 Report

The work by Xu, Tsai, and Palke titled, "Study of 405 nm laser-induced time-resolved Photoluminescence spectroscopy on spinel and alexandrite." is an interesting use of a relatively straightforward and inexpensive technique to differentiate various gemstones. The work is deserving to be published, however, improvements to the overall quality of the presentation should be made. Below are suggestions and recommendations to enhance the appeal of the work. 

1. In the title, the "P" of photoluminescence is capitalized where it should not be and there should not be a period "." at the end.

2. In the list of keywords, there are numbers after each keyword, these should be removed.

3. Throughout the manuscript there are inconsistencies in formatting with the subscripts and super scripts. Especially the oxidation states should be as superscript, e.g. Al3+ and Cr3+ in lines 37, 42 and the t1/2 should be subscript for instance in the header of Table 2 and line 239 as well as other places I might have missed. 

4. Line 65, "radiation transition" should be "radiative transition".

5.    The figures have different formatting which cheapens the quality feeling of the work. For instance, all the font sizes and display sizes of the figures should be consistent. The use of square brackets [ ] versus parentheses ( ) for units should be consistent, the x-axes of Figures 4, 8, and 9 have strange units of 10 nanoseconds and large values in the 1000s for the time -- please pick more natural and intuitive numbers to minimize the confusion this causes.

6. Line 85 discusses the 1/e time (i.e. the tau in the exponential function) but then the rest of the manuscript refers to the half-life: ln(2)*tau making this discussion seem out of place. Please make that consistent. 

7. "emmison" is misspelled on line 201. 

8. Lastly, seeing as this is a relatively uncommon way of doing time-resolved photoluminescence measurements, the data would be more convincing with a direct measurement of the instrument response to verify that these measurements are not all limited by the sampling interval as is the case in the short-time range of figure 9.

Author Response

  1. In the title, the "P" of photoluminescence is capitalized where it should not be and there should not be a period "." at the end.

Response 1: Thank you for bringing this to our attention. We have modified capitalization of photoluminescence and removed the period.

  1. In the list of keywords, there are numbers after each keyword, these should be removed.

Response 2: Thank you for your feedback on the list of keywords. We have removed them in the final version of the manuscript.

  1. Throughout the manuscript there are inconsistencies in formatting with the subscripts and super scripts. Especially the oxidation states should be as superscript, e.g. Al3+ and Cr3+ in lines 37, 42 and the t1/2 should be subscript for instance in the header of Table 2 and line 239 as well as other places I might have missed. 

Response 3: Thank you for your helpful feedback on the formatting of subscripts and superscripts in our manuscript. We will carefully review the manuscript and make sure to correct all instances where the formatting is incorrect.

  1. Line 65, "radiation transition" should be "radiative transition".

Response 4: Thank you for your feedback. We have modified the term "radiative transition"   instead of "radiation transition".

  1.   The figures have different formatting which cheapens the quality feeling of the work. For instance, all the font sizes and display sizes of the figures should be consistent. The use of square brackets [ ] versus parentheses ( ) for units should be consistent, the x-axes of Figures 4, 8, and 9 have strange units of 10 nanoseconds and large values in the 1000s for the time -- please pick more natural and intuitive numbers to minimize the confusion this causes.

Response 5: Thank you for your feedback on the figures in our manuscript. We will carefully review and revise all the figures to ensure that the font sizes and display sizes are consistent, and that the units are presented in a consistent manner.  Regarding the x-axes of Figures 4, 8, and 9, we agree that the current units of 10 nanoseconds and large values may be confusing, and we will modify them into unit of ms in the final version of the manuscript.

  1. Line 85 discusses the 1/e time (i.e. the tau in the exponential function) but then the rest of the manuscript refers to the half-life: ln(2)*tau making this discussion seem out of place. Please make that consistent. 

Response 6: Thank you for your feedback. We want to clarify that the lifetime of photoluminescence is the time measured for the excited energy to decay exponentially to N/e (36.8%) of the original, regardless of whether it is one-phase or multi-phase exponential decay. In this study, we used the half-life t1/2 as an exponential parameter to characterize the decay curve, alternatively is to use τ: decay constant, however we think half-life t1/2 is still easier for readers to understand, which is indirectly related to the lifetime. We will make sure to use the correct terminology consistently throughout the manuscript and clarify this point in the final version. Thank you again for your helpful feedback.

  1. "emmison" is misspelled on line 201. 

Response 7: Thank you for your feedback. We will correct this mistake in the final version of the manuscript to ensure the accuracy and clarity of our writing.

  1. Lastly, seeing as this is a relatively uncommon way of doing time-resolved photoluminescence measurements, the data would be more convincing with a direct measurement of the instrument response to verify that these measurements are not all limited by the sampling interval as is the case in the short-time range of figure 9.

Response 8: Thank you for your suggestion. We agree with you that the proposed design is not the ideal setup for time-resolved photoluminescence spectroscopy. For optimal results, an ideal setup would either employ a time-gated detector or Time-Correlated Single Photon Counting (TCSPC) for short-time scale PL response. However, the proposed method is sufficient to distinguish between natural and synthetic spinels and alexandrites, as their photoluminescence decay times are typically between 10 to 50 ms.

In order to clarify the limitation, a paragraph will be added in line 290 – 292 (302-304) in Section 3.2.2.: We note that due to the minimal sampling interval of our system, photoluminescence lifetimes shorter than 5 ms may not be confidently measured. Based on our measurement, the B2 site decay time is approximately 2 ms.

Round 2

Reviewer 2 Report

manuscript can be recommended for publication